# Fabry Disease: Molecular Basis, Pathophysiology, Diagnostics and Potential Therapeutic Directions

**DOI:** 10.3390/biom11020271

**Published:** 2021-02-12

**Authors:** Ken Kok, Kimberley C. Zwiers, Rolf G. Boot, Hermen S. Overkleeft, Johannes M. F. G. Aerts, Marta Artola

**Affiliations:** 1Department of Medical Biochemistry, Leiden Institute of Chemistry, Leiden University, P.O. Box 9502, 2300 RA Leiden, The Netherlands; k.kok@lic.leidenuniv.nl (K.K.); k.c.zwiers@lic.leidenuniv.nl (K.C.Z.); r.g.boot@LIC.leidenuniv.nl (R.G.B.); 2Department of Bio-organic Synthesis, Leiden Institute of Chemistry, Leiden University, P.O. Box 9502, 2300 RA Leiden, The Netherlands; h.s.overkleeft@chem.leidenuniv.nl

**Keywords:** lysosomal storage disorders, Fabry disease, α-galactosidase A, A4GALT, globotriaosylceramide (Gb3), globotriaosyl-sphingosine (lysoGb3), enzyme replacement therapy, pharmacological chaperone therapy, substrate reduction therapy

## Abstract

Fabry disease (FD) is a lysosomal storage disorder (LSD) characterized by the deficiency of α-galactosidase A (α-GalA) and the consequent accumulation of toxic metabolites such as globotriaosylceramide (Gb3) and globotriaosylsphingosine (lysoGb3). Early diagnosis and appropriate timely treatment of FD patients are crucial to prevent tissue damage and organ failure which no treatment can reverse. LSDs might profit from four main therapeutic strategies, but hitherto there is no cure. Among the therapeutic possibilities are intravenous administered enzyme replacement therapy (ERT), oral pharmacological chaperone therapy (PCT) or enzyme stabilizers, substrate reduction therapy (SRT) and the more recent gene/RNA therapy. Unfortunately, FD patients can only benefit from ERT and, since 2016, PCT, both always combined with supportive adjunctive and preventive therapies to clinically manage FD-related chronic renal, cardiac and neurological complications. Gene therapy for FD is currently studied and further strategies such as substrate reduction therapy (SRT) and novel PCTs are under investigation. In this review, we discuss the molecular basis of FD, the pathophysiology and diagnostic procedures, together with the current treatments and potential therapeutic avenues that FD patients could benefit from in the future.

## 1. Introduction

In 1898, two dermatologists, Johannes Fabry in Dortmund and William Anderson in London, reported similar patients with characteristic skin lesions, so-called angiokeratoma corporis diffusum [1,2]. The inherited disorder became known as Anderson–Fabry disease, nowadays generally referred to as Fabry disease (FD). A striking feature of FD (OMIM 301500) is the characteristic lipid deposits, named zebrabodies, prominently encountered in endothelial cells but lesser also in other cell types [3]. The main component of the storage material was identified by Sweeley and Klionsky as the globoside globotriaosylceramide (Gb3), initially named ceramidetrihexoside (CTH) [4]. Additional accumulating glycosphingolipids in FD patients such as galabiosylceramide (Gb2) and blood group B, B1 and P^1^ antigens can be observed, all sharing a terminal α-galactosyl moiety [3]. The molecular basis for lipid abnormalities was firstly elucidated by Brady and coworkers, demonstrating the deficiency of lysosomal acid α-galactosidase activity converting Gb3 to lactosylceramide (LacCer) [5]. A convenient enzyme assay for diagnosis of FD was next developed by Kint, employing an artificial chromogenic α-galactoside substrate [6]. Subsequent research revealed that the reduced α-galactosidase activity in FD patients stems from the lysosomal enzyme α-galactosidase A (α-GalA) that is encoded by the *GLA* gene located at chromosome Xq22 [3]. Of note, an ancient gene duplication has led to two relatively homologous genes: *GLA* and *NAGA*. *NAGA* (also known as α-Galactosidase B (α-GalB)), locus 22q13.2, evolved into an *N*-acetylgalactosaminidase cleaving α-*N*-acetylgalactosamine from glycoconjugates [7,8]. Mutations in NAGA cause Schindler disease and Kanzaki disease [9]. α-GalA and α-GalB are both inhibited in enzymatic activity by galactose but only α-GalB is inhibited by *N*-acetylgalactosamine [10]. The globoside Gb3 is degraded by α-GalA, although a minor α-GalB activity towards this metabolite has been reported [11]. The α-GalA enzyme is synthesized as a 429 aa precursor that is processed to a 398 aa glycoprotein functioning as a homodimer [11,12]. The three N-linked glycans of α-GalA acquire mannose-6-phosphate moieties that assist the enzyme’s sorting to lysosomes by mannose-6-phosphate receptors. The activity of α-GalA towards the lipid substrate is enhanced by the activator protein saposin B and negatively charged lipids [3]. Close to 1000 mutations have meanwhile been identified in the *GLA* gene, of which most are missense mutations. Thanks to the work of many, particularly by Sakuraba and colleagues, the consequences at the enzyme level of several α-GalA mutations are known [13]. However, the impact of a large number of the presently reported α-GalA mutations remains unclear [14]. So-called α-GalA mutations of unknown significance are often not associated with clearly reduced α-galactosidase activity, promoting the debate as to whether they truly are causing FD [15].

## 2. Clinical Manifestation of FD

The classic disease manifestation of FD has been extensively described for males [3,16]. Generally, these FD hemizygotes show α-GalA mutations with no or very little residual α-galactosidase activity. Besides the characteristic angiokeratoma, the patients develop corneal opacity (cornea verticillata), neuropathic pain (acroparasthesias), intolerance to heat, inability to sweat, micro-albuminuria and increased intima media thickness. Later in life, the patients develop progressive kidney disease, cardiac symptoms and cerebrovascular disease (stroke). These late-onset symptoms are indistinguishable from similar complications of other origin commonly occurring in the general population. The renal disease usually involves progressive proteinuria following a decline in the glomerular filtration rate (GFR). The final outcome is end-stage renal disease requiring dialysis and kidney transplantation. The heterogeneous cardiac complications may include progressive hypertrophic cardiomyopathy, conduction defects and arrhythmia, atrial fibrillation, valvular disease and coronary artery stenosis. Regarding cerebrovascular complications, ischemic stroke and transient attacks occur relatively commonly. Brain MRI often reveals asymptomatic lesions in the white matter [3].

It has only more recently been appreciated that a significant portion of female FD heterozygotes develop complications, although usually in an attenuated form compared to male FD hemizygotes [17]. Due to X chromosome inactivation (Lyonization), wherein there is (random) transcriptional silencing of one of the X chromosomes in each cell, FD females are mosaic for the expression of α-GalA. Skewed X-inactivation favoring the mutant α-galactosidase A allele in female FD heterozygotes is associated with more severe disease manifestation [18]. In FD females, chronic renal insufficiency is rare. The manifestation of symptoms in FD females is remarkable given the known mosaic of α-GalA-containing and α-GalA-deficient cells in their tissues and the considerable levels of active α-GalA in the circulation [3]. Of note, heterozygous carriers of another X-linked lysosomal storage disorder (LSD) caused by iduronate 2-sulphatase deficiency, Hunter disease (HD), lack symptoms [19]. Apparently, in the case of HD, but not FD, sufficient complementation occurs in deficient cells of female heterozygotes owing to the uptake of secreted enzymes by normal cells [20]. Importantly, atypical variants of FD have been recognized. In these individuals, the disease is restricted to a single organ, particularly the heart and kidneys [21]. FD is now considered to be the most common LSD [16,22]. An accurate estimation of the prevalence is complicated by the great phenotypic heterogeneity. The estimated birth prevalence of classic FD is 0.42 per 100,000 male births in the Netherlands. The actual total prevalence of FD is higher because of under diagnosis of female patients and atypical disease manifestations. Newborn screening studies based on identification of abnormalities in the *GLA* gene or deficiency in α-GalA activity suggest a birth prevalence of at least 1 in 4000 in European populations [23], and higher frequencies have even been noted in Taiwan [24].

## 3. Storage Cells and Secondary Storage Lipids

The clinical symptoms and signs of FD differ fundamentally from other sphingolipidoses such as Gaucher disease (GD), in which lipid-laden macrophages are prominent and thought to contribute to characteristic symptoms such as hepatosplenomegaly and pancytopenia [25]. In sharp contrast, multiple cell types accumulate lipids in classic FD patients [26]. In the kidney, for example, lipid deposits are detected by electron microscopy in podocytes, endothelial glomerular cells and distal tubular cells [27]. Another peculiarity of FD is the relative mild outcome of complete α-GalA deficiency encountered in most classic FD males. There are no infantile and severe juvenile FD phenotypes as observed for other sphingolipidoses. Moreover, in FD, there is a remarkable discrepancy between the onset of lipid storage and that of symptoms. Gb3 storage in classic FD males already occurs in utero in endothelial cells and macrophages [28]. However, clinical symptoms develop only late in life. The same discrepancy is noted in α-GalA-deficient FD mice and rats [29,30]. Lipid-laden macrophages have been observed in the liver of classic FD males. Consistent with this, chitotriosidase, an established plasma biomarker of sphingolipid-accumulating macrophages in GD patients, is also elevated in the plasma of classic FD males [31,32]. This is not the case in FD females, suggesting that their macrophages are complemented by enzymes released from surrounding cells in contrast to other cell types [33].

A hallmark of FD is the marked elevation of water-soluble deacylated Gb3, also known as globotriaosyl-sphingosine (lysoGb3) [34]. The sphingoid base lysoGb3 is formed by the enzyme acid ceramidase from accumulating Gb3 in lysosomes [35]. LysoGb3 can leave cells and reach the circulation, resulting in over a hundred-fold elevated plasma levels in classic FD males. LysoGb3 is even clearly raised in the plasma of many female FD heterozygotes. Increases in lysoGb3 were also observed in the urine of FD patients. A similar lysoGb3 abnormality was detected in FD mice [36]. Several investigators have meanwhile confirmed the value of elevated plasma lysoGb3 as a biomarker of classic FD, including demonstration of abnormal lysoGb3 in urine [37,38,39,40]. Prominent sources of plasma lysoGb3 are likely the endothelium and liver, and the increased plasma lipid appears not to reflect one particular symptom [25,41].

## 4. Pathophysiology

It is well established that accumulation of Gb3 during α-GalA deficiency takes place in lysosomes, but the subsequent mechanisms causing cellular dysfunction, and ultimately symptoms, are still poorly understood [25,42]. As with other inherited glycosphingolipidoses, lipid-laden lysosomes can be envisioned to cause impaired autophagic flux, including mitophagy, contributing to the observed mitochondrial dysfunction in fibroblasts of FD patients [43,44,45]. Likewise, dysfunction of the endoplasmic reticulum may occur as suggested by the observed induction of the unfolded protein response in cells of some FD patients [46]. Fibrosis, inflammation and oxidative stress seem to play key roles in pathogenesis [47,48,49,50]. It has been hypothesized that lysoGb3 may also act as a pathogenic factor in FD [25,51]. A significant correlation of lysoGb3 lifetime exposure with overall disease severity was noted for classic male and female FD patients [41]. Indeed, lysoGb3 promotes smooth muscle cell proliferation, which fits with the increased intima media thickness and arterial stiffness in FD [29]. Furthermore, evidence has been provided that lysoGb3 at concentrations occurring in FD males damages nociceptive neurons, consistent with the reported pain in the extremities of classic FD males [52]. Lifetime exposure to lysoGb3 was found to correlate very significantly with the cold detection threshold and thermal sensory limen of the upper limb [53]. Next, lysoGb3 is thought to contribute to podocyte loss and glomerulus fibrosis, important aspects of the renal disease in FD patients [54,55]. Finally, lysoGb3, at concentrations as in FD patients, is found to inhibit endothelial nitric oxide synthase (eNOS) and thus may contribute to the vasculopathy in FD [56,57].

There appear to be other cellular consequences of α-GalA deficiency beyond the lysosome. The autophagy–lysosome pathway (ALP) is an important recycling pathway that mediates cell survival [58]. Disruption of the ALP is a common hallmark of lysosomal storage disorders, including Fabry disorders [59,60,61]. Likewise, in sphingolipid disorders such as Gaucher disease and Fabry disease, disturbed mitochondrial function and energy balance have been noted (for an excellent review on this topic, see Ivanova et al. 2020) [45]. Moreover, infiltration of lymphocytes and macrophages in tissues of FD, including the heart, has been observed, suggesting a role for inflammation in tissue damage. Possibly, chronic inflammation in FD, and associated oxidative stress, promotes organ damage (for a review, see Rozenfeld et al. [48]).

## 5. Diagnosis

Monitoring of disease manifestations and therapeutic efficacy of FD treatment is essential for the clinical management of FD patients. Disease onset and progression can be determined by clinical, radiological and laboratory analysis. However, the efficacy of a clinical treatment is sometimes challenging to assess due to high variability among patients. In addition, some pathological consequences of FD such as advanced renal failure are irreversible. Nevertheless, biomarkers play a very important role in disease and treatment monitorization [62].

The diagnosis of classic FD males is straightforward: identification of *GLA* gene mutations encoding an absent or evidently dysfunctional α-GalA protein. Extremely low α-GalA activity in leukocytes, fibroblasts and dried blood spots can be conveniently demonstrated using artificial water-soluble substrates, such as 4-methylumbelliferyl-α-galactoside [25]. Detection of elevated concentrations of plasma and urinary Gb3 and lysoGb3 can be used to further confirm diagnosis [29,37,62]. Sensitive LC-MS methods for this have been developed [63,64,65,66]. Enzyme activity assays are not always informative for FD females, particularly those with unfavorably skewed X-inactivation. Detection of elevated lysoGb3 is very helpful then to confirm FD diagnosis in females. Problematic is the diagnosis of atypical FD patients presenting with an uncharacteristic symptom (e.g., albuminuria, left ventricular hypertrophy or white matter lesions) in combination with an abnormality in the *GLA* gene with unknown consequences. This is often accompanied by a relatively high residual enzyme activity in cells and no clear abnormality in plasma or urinary Gb3 and lysoGb3 concentrations. Analysis of biopsies and demonstration of deposits of Gb3 are considered, in problematic cases, as helpful to support diagnosis [67,68]. Biochemical monitoring of disease in FD patients increasingly relies on the measurement of plasma lysoGb3; however, it should be kept in mind that the lipid levels do not reflect a particular symptom [69].

## 6. α-GalA: Reaction Mechanism and Activity-Based Probes (ABPs)

Glycosidases are hydrolytic enzymes that ensure the cleavage of glycosidic linkages in (oligo) saccharides and glycoconjugates, and they have been essential for the breakdown of various glyco(sphingo)lipids such as globotriaosylceramide (Gb3), which together with lysoGb3 is the predominant glycosphingolipid that accumulates in FD patients. α-GalA, an exo-retaining galactosidase member of the GH27 family (cazypedia.org), is responsible for the breakdown of Gb3 into lactosylceramide (LacCer) and galactose [70]. α-GalA cleaves the terminal α-linked galactose units from polysaccharides, glycolipids and glycoproteins [5]. α-GalA and its related lysosomal counterpart α-*N*-acetylgalactosaminidase (NAGA), also known as α-galactosidase B (α-GalB), which cleaves terminal α-linked *N*-acetylgalactosamine (α-GalNAc) moieties, are the only human retaining α-galactosidases known [71]. Their active sites only differ in two amino acid residues, which accommodate the C-2 substituent of the enzymatic substrate [12,71]. α-GalB can accommodate larger C-2 substituents (Figure 1A) in contrast to α-GalA (Figure 1B) which only allows a secondary hydroxyl at the C-2 position. Interestingly, by changing these two active site residues in either enzyme, the substrate specificity can be interchanged [71]. In addition to the C-2 position, the substituent and conformation at the C-6 position also play an important role in determining the reactivity of carbohydrates and the selectivity of chemical glycosylation reactions by influencing the stability of the oxocarbenium ion [72]. Most side chains of unbound “free” sugar molecules populate either a *gauche,gauche* (*gg*), *gauche,trans* (*gt*) or *trans*,*gauche* (*tg*) conformation, in which these abbreviations refer to the stereochemical relation between the O6-C6, O5-C5 and C4-C5 bonds. This results in the *gg* conformation being the most favorable for the formation of oxocarbenium ions [72]. However, this is highly influenced by the stereochemistry of the substituent at the C4 position which is reflected by the fact that the C6 side chains of galactose-configured molecules tend to adopt the *gt* conformation since the *gg* conformation results in an energy penalty due to both C6 and C4 substituents having an axial orientation. Interestingly, these stereochemical preferences are also reflected in the way glycosidases bind their substrates. In the case of α-galactosidases, they have a preference for binding their substrates in the *gt* conformation, thereby avoiding additional energy penalties but still maintaining the highly stabilizing effect on the oxocarbenium ion transition state that the substrate adopts during hydrolysis [72].

Apart from the structural similarities between α-GalA and α-GalB, both enzymes retain galactosidases, which means that cleavage of the glycosidic linkage in the enzymatic substrate results in retention of the stereochemistry at the anomeric position of the terminal galactose moiety [73]. This retention of stereochemistry at the anomeric position is driven by a Koshland double displacement mechanism (Figure 1C) [74,75]. In the first step of this mechanism, the nucleophilic residue attacks at the anomeric center of the substrate, while the acid/base residue protonates the leaving group (LacCer in the case of α-GalA). This first step results in the formation of a covalent intermediate via an oxocarbenium ion-like transition state [75]. For the second step, the acid/base amino acid deprotonates a water molecule which concomitantly performs a nucleophilic attack at the anomeric center of the covalently bound substrate. This second step also follows a second oxocarbenium ion-like transition state which results in hydrolysis of the substrate with net retention of the stereochemistry at the anomeric position. X-ray crystal structures revealed that the roles of the nucleophile and the catalytic acid/base in α-GalA were performed by aspartic acid residues D170 and D231, respectively [12]. In addition, the crystal structure showed that α-GalA is a homodimeric glycoprotein and each of the monomers contains two domains, one active site domain and one C-terminal domain containing eight antiparallel β-strands and two sheets. Furthermore, it was found that each monomer contains three *N-*glycosylation sites that are important for the transport of the enzyme towards the lysosome mediated by the mannose-6-phosphate receptor [12].

Due to their medical implications, multiple inhibitors have been developed over the years that affect both α-galactosidases and more specifically α-GalA. These inhibitors can generally be divided into reversible or irreversible inhibitors. Irreversible inhibitors bind covalently to the enzyme, thereby capitalizing on the Koshland double displacement mechanism. Thus, these inhibitors often utilize an electrophilic trap to capture the nucleophilic residue in the enzyme active site. Some of the oldest irreversible glycosidase inhibitors, which were also designed for α-galactosidases, are fluorinated sugars such as **1** [76] (Figure 2). The fluorine atom causes an inductive effect, which makes it more difficult for the substrate to enter the positively charged oxocarbenium ion transition state and impairs its hydrolysis. Unfortunately, these fluorinated sugars showed modest to no inhibition of α-galactosidases from green coffee bean and *Aspergillus niger*. Epoxides are commonly used electrophilic traps and they were first used as α-galactosidase inhibitors in the form of conduritol C **2** [77]. This epoxide-based inhibitor was further developed into the synthetic form of the cyclophellitol epoxide **3** [78,79]. Unfortunately, epoxide **3** is not a selective α-GalA inhibitor since it also inhibits β-galactosidases GLB1 and GALC [79]. In addition to the epoxides, their nitrogen-based counterpart α-galactose-configured aziridine **4** [78] has shown to be a potent inhibitor of α-GalA (apparent IC_50_ = 40 nM) [79]. However, similar to epoxide **3,** the aziridine is not selective for α-GalA and displays a decent inhibition of GLB1 (apparent IC_50_ = 0.93 µM) and GALC (apparent IC_50_ = 1.1 μM). Both epoxide- and aziridine-based inhibitors make use of the ^4^*C*_1_
→
^4^*H*_3_
→
^1^*S*_3_ conformational itineraries of retaining α-galactosidases [75], mimicking the ^4^*H*_3_ transition state which is also adopted by β-galactosidases [80]. Recently, α-Gal-cyclosulfate **5** has been synthesized as a potential α-GalA inhibitor which mimics the initial ^4^*C*_1_ Michaelis complex [79]. Its chair conformation may render this inhibitor selective towards α-GalA (apparent IC_50_ = 25 μM) and binds covalently to the enzyme adopting a ^1^*S*_3_ bound conformation [79].

Next to these covalent compounds, one of the first α-GalA inhibitors that is currently used in the clinic is the reversible inhibitor 1-deoxy-galactonojirimycin (DGJ, **6**) [81] which exploits non-covalent interactions. The endocyclic nitrogen of **6** can become protonated, forming an ion pair with a negatively charged amino acid residue in the α-GalA active site. Iminosugar **6** is a potent α-GalA inhibitor (IC_50_ = 79 nM) but lacks selectivity since it also inhibits both GLB1 and β-glucosidase GBA [79]. As a potential alternative for **7,** selective cyclosulfamidate **7** was designed [79]. The cyclosulfamidate is a reversible inhibitor that results from the replacement of one of the endocyclic oxygens of the cyclosulfate **5** by a nitrogen atom. This replacement severely decreases its leaving group capacity, turning cyclosulfamidate **7** into a reversible inhibitor mimicking the Michaelis complex conformation. Although cyclosulfamidate **7** is a more selective inhibitor, it presents a lower inhibitory potency than iminosugar **6**.

Apart from their application as irreversible inhibitors, epoxide- and aziridine-based inhibitors have also been functionalized into activity-based probes (ABPs) for the labeling of various glycosidases in biochemical assays [80]. Modification of aziridine **4** with acyl-based fluorophores (**8** and **9**) and biotin (**10**) tags results in valuable biochemical tools to study α-GalA [82]. These ABPs have shown great selectivity towards α-GalA and α-GalB and can be used in competitive activity-based protein profiling (cABPP) assays to screen new inhibitors or to profile enzyme activity in cell extracts. In particular, ABPs **8** and **10** have been used to study the activity of α-GalA and α-GalB from plant extracts to study the potential enzyme production from *Nicotiana Benthamiana* (*N. Benthamiana*) [10]. In addition, these probes have also been used to identify a novel α-galactosidase from *N. Benthamiana* named α-galactosidase A1.1 [83]. This plant-derived enzyme presents significant structural similarities with α-GalA, an improved stability over a broad pH range and a similar ability to hydrolyze both Gb3 and LysoGb3, representing a potential therapeutic alternative for ERT-based FD management.

## 7. Present α-GalA-Centered Therapy Approaches

Until a few decades ago, there was no effective treatment available for inherited lysosomal storage diseases (LSDs). The management of most LSDs consisted only of supportive care. For some of the disorders, particularly mucopolysaccharidosis I-H (MPS I-H) and globoid cell leukodystrophy, bone marrow transplantation was performed [84,85]. A breakthrough regarding treatment of LSDs was accomplished by Roscoe Brady, who pioneered, with colleagues at the National Institutes of Health (NIH) in Bethesda, the development of an effective enzyme supplementation for non-neuronopathic type 1 Gaucher disease patients [86,87]. Currently, LSDs treatment capitalizes on four main therapeutic strategies (Figure 3). Intravenous supplementation of administered enzyme replacement therapy (ERT) increases the enzyme levels in the body, while oral pharmacological chaperones (PCT) have shown to promote the correct folding of amenable mutated glycosidases and retrieve residual activity levels. Substrate reduction therapy (SRT) aims to inhibit the biosynthesis of the accumulated metabolites. More recently, gene/RNA therapy allows the insertion of the gene encoding the deficient enzyme in patient cells. Unfortunately, FD patients can presently benefit only from ERT and, since 2016, PCT, both always combined with supportive adjunctive and preventive care.

Enzyme replacement therapy (ERT) is based on chronic two-weekly infusion of (now recombinant) glucocerebrosidase targeted to macrophages by the presence of terminal mannose residues in its N-linked glycans (Figure 3A). The success of the intervention prompted the development of similar ERT approaches for other LSDs, including FD. For this purpose, two different recombinant α-GalA preparations were independently developed in academic centers and subsequently pharmaceutical companies [88,89]. On 3 August 2001, both enzymes for ERT of FD were approved as the first orphan drugs in Europe: agalsidase alfa (Replagal^®^, Shire HGT [90]) and agalsidase beta (Fabrazyme^®^, Sanofi Genzyme [90]). The production of the two enzymes is fundamentally different: agalsidase alfa is produced by gene promotor activation in fibroblasts and agalsidase beta by conventional cDNA technology in CHO cells. This difference seemed highly relevant since mRNA editing had been reported for α-GalA. Theoretically, mRNA editing of agalsidase alfa, but not of agalsidase beta, would cause an amino acid difference at position 396 of both enzymes. Detailed analysis of the amino acid composition of both enzymes revealed that α-GalA mRNA is not edited [91]. Both recombinant enzymes, differing slightly in glycan composition, were found to be comparable when tested on in vitro specific activity and uptake by cultured fibroblasts [91,92]. Recent studies with different cultured cells revealed that uptake of recombinant α-GalA (clathrin- and caveolae-dependent endocytosis) might be cell type-specific [93]. Of note, in a human podocyte cell line, three endocytic receptors, IGF2R/M6P, megalin and sortilin, were reported to be involved in α-Gal A uptake [94]. Attention has been focused on improving the tissue distribution of therapeutic enzymes by the generation of α-GalA glycoforms. Elegant chemoenzymatic synthesis was employed by Fairbanks and colleagues to replace the glycans of recombinant α-GalA by synthetized mannose-6-phosphate-rich structures [95]. More recently, engineered CHO cell lines were used to generate specific α-GalA glycoforms [96]. Bolus injection in FD mice revealed the impact of glycan composition on the biodistribution of α-GalA. Unexpectedly, an α-2-3 sialylated (SA) glycoform of α-GalA was found to exhibit improved circulation and biodistribution [96]. It should, however, be kept in mind that translating the outcome of a bolus injection administered via the tail vein in mice to the biodistribution in FD patients is tricky.

Both agalsidases alfa and beta are now approved in many countries throughout the world, but agalsidase alfa is still not approved by the US Food and Drug Administration (FDA). A recombinant α-GalA named Pegunigalsidase-alfa (PRX-102, prh-α-GalA from Protalix) that is produced in tobacco cells and has been chemically modified with polyethyleneglycol (PEG) is currently being investigated in clinical trials [97]. Such chemical modification offers protein stabilization, increased half-life and an improved biodistribution profile [98]. The registered ERTs (agalsidase beta at a standard dose of 1 mg/kg bw/2 wks and agalsidase alfa at a standard dose of 0.2 mg/kg bw/2 wks, similar in costs) were both found to result in clearance of storage material in heart and kidney biopsies. Based on this, ERT was hoped to protect kidney and cardiac function, but more recent data indicate that new clinical events (such as development of end-stage renal failure, myocardial infarction, ventricular fibrillation or cerebrovascular events) may occur in FD patients during ERT. Male sex, classical phenotype and increasing age at treatment initiation are risk factors for progression of disease while on ERT. Other risk factors are reduced renal function, proteinuria, cardiac hypertrophy and fibrosis, hypertension and occurrence of events before the start of ERT. An earlier start of ERT, especially in male patients with classic FD, is thought to improve the treatment outcome [68,99].

The impact of ERT on plasma lysoGb3 levels in FD patients has been, and still is, widely monitored [25]. Plasma lysoGb3 in classic FD patients was found to decrease rapidly after the start of ERT with several regimens in an enzyme dose-dependent manner [100]. After 3 months of treatment, plasma lysoGb3 levels tended to become stable but complete corrections were rare. On the other hand, a reduction in ERT was found to lead to increases in Gb3/lysoGb3 levels in most FD patients investigated [101]. Some classic FD males showed, after a few months of ERT at a similar enzyme dose, a relapse in plasma lysoGb3 levels, which prompted the analysis of a possible antibody response to the therapeutic enzyme. Indeed, the occurrence of antibodies is observed in about 70% of classic FD males receiving ERT [102,103]. Most classic FD males completely lack the α-GalA protein and an immunological response to the infused foreign therapeutic protein is not surprising. The antibodies formed in classic FD male patients receiving agalsidase alfa or beta comparably bind to both recombinant enzymes in vitro and neutralize enzyme activity in vitro [103]. The correction of plasma lysoGb3 during ERT is much less prominent in FD males with antibodies than those without [103]. Similarly, urinary Gb3 levels also hardly correct in FD males with antibodies [103]. The clinical consequences of neutralizing antibodies were unclear for many years. Bénichou et al. observed significantly impaired Gb3 clearance in skin biopsies of patients treated with ERT showing high antibody titers [104]. A five-year study with 68 male FD patients treated with ERT showed that 40% presented serum-mediated antibody inhibition of enzyme, which was associated with increased lyso-Gb3, higher risks for FD-associated symptoms and impaired cardiac and renal function [25]. The cause(s) for the limited response to ERT is (are) not known. Likely, the induction of (neutralizing) antibodies against a therapeutic protein in classic FD males contributes to this. Inadequate ERT biodistribution has also been highlighted, with few enzymes reaching podocytes and cardiac myocytes [28,29].

A seminal work by Ishii and colleagues revealed that specific mutant forms of α-GalA that misfold in the endoplasmic reticulum and are subsequently prematurely degraded can be partly rescued by galactose and more potently by 1-deoxygalactonojirimycin [105,106,107]. These findings prompted the development of 1-deoxygalactonojirimycin as a pharmacological chaperone named Migalastat (Galafold^®^, Amicus Therapeutics), which was approved in 2016 in Europe and Canada (USA approval was delayed to 2018) as an alternative therapeutic approach, representing the only oral treatment for FD (Figure 3B) [108]. This small iminosugar reversibly binds to the enzymatic active site in the endoplasmic reticulum (ER) and stabilizes, at low concentrations, particular mutant forms of α-GalA (known as amenable mutant forms), promoting the proper folding of the enzyme, maturation and its trafficking to lysosomes [106,109]. The more acidic lysosomal pH (compared with a neutral pH in the ER) and the high concentration of the Gb3 metabolite in the lysosome displace the reversible small chaperone from the active site and the active enzyme is then able to hydrolyze the accumulated substrates at the lysosomal interface. However, this therapeutic strategy is limited to a specific number of mutations. It is estimated that only 35–50% of FD patients present a migalastat-amenable mutation [110,111]. Interestingly, several recent studies on FD patients with amenable mutations suggest that switching from ERT with agalsidase alfa or beta to migalastat can be a valid, safe and well-tolerated strategy [111,112,113]. A more recent strategy involves the joint administration of a recombinant enzyme and a pharmacological chaperone [112,113,114,115], aiming to stabilize the recombinant enzyme in circulation with the final goal of increasing the concentration of the enzyme that may reach the affected tissues, and allowing the use of lower enzyme doses and prolonged intervals between IV administrations, which should ultimately decrease the side effects and treatment cost and, more importantly, improve the quality of life of FD patients [116]. In particular, migalastat presents a synergetic effect in cultured fibroblasts from FD patients and increases the tissue uptake of recombinant human α-GalA in FD mice [113,115]. A conceptually new class of enzyme stabilizers, cyclophellitol cyclosulfamidates, have recently been described to stabilize algasidase beta and increase α-GalA activity in FD fibroblasts, assisting the functional correction of lysoGb3 metabolite accumulation [79]. Positive allosteric modulators are also under investigation which could afford safer daily dose regimens by avoiding the use of active site binders with a potential inhibitory effect. In particular, in silico docking leads to the identification of 2,6-dithiopurine, an allosteric ligand that stabilizes lysosomal α-GalA in vitro and rescues a particular mutant form, A230T, which is a non-amenable mutation for PCT 1-deoxygalactonojirimycin [117].

Extensive research efforts have been made towards a better FD therapy over the past twenty years, and clinical trials have resulted in FDA- and/or EMA-approved ERT and PCT. Importantly, new research in the field moves towards SRT and gene/RNA therapy to fill the gap of this yet not curable disease. Substrate reduction therapy (SRT) relies on small molecules capable of inhibiting the biosynthesis of the metabolites that accumulate in the lysosome (Figure 3C). SRT using GlcCer synthase (GCS) inhibitors such as miglustat and eliglustat is already on the market for the treatment of Gaucher disease type I, and miglustat is approved for Niemann–Pick Type C, a rare progressive genetic disorder characterized by the deficient transport of cholesterol and lipids inside cells. Both drugs inhibit GCS, which blocks the first step in glycosphingolipid biosynthesis [118]. In particular, venglustat/ibiglustat [119] and lucerastat [120] are currently under evaluation as oral GCS inhibitors for FD (NCT02228460 and NCT02930655 are the respective clinical trials). Efforts and directions towards SRT for FD with specific inhibition of A4GALT, the responsible glycosyltransferase for the synthesis of Gb3, will be further discussed in this review.

Gene therapy is based on the insertion of a correcting gene, encoding the deficient enzyme, in patient cells (Figure 3D). The correcting gene is usually delivered through a vector such as adeno-associated virus (AAV), lentivirus, retrovirus or a non-viral-based system that can then alter the DNA or RNA transcript used for the synthesis of the enzyme of interest. By inserting the nonmutant *GLA* gene, gene therapy aims to correct the enzyme deficiency and reduce the accumulation of Gb3 and lysoGb3 and eventually prevent organ damage in FD patients. A first-in-human clinical study for the treatment with autologous stem cell transplantation using CD34+ cells transduced with the lentiviral vector containing the human *GLA* gene started in Canada in 2016 (NCT02800070). Avobrio is also currently running a phase II clinical trial (NCT03454893) to study the efficacy and safety of a gene therapy (AVR-RD-01) for the treatment of classic FD patients. Recently, two new gene therapies (ST-920 and FLT190) making use of an AAV vector encoding human α-GalA cDNA with specific liver expression cassettes have been described to increase plasma and tissue α-GalA activities in an FD mouse model and are in phase I/II clinical trials (NCT04046224 and NCT04040049) [121,122,123]. Messenger RNA (mRNA) is also emerging as a new class of therapy for the treatment of rare monogenic disorders. In particular, the efficacy of a messenger mRNA encoding the α-GalA enzyme has been reported in FD α-GalA knockout mice through an IV bolus administration of α-GalA mRNA encapsulated in lipid nanoparticles (0.05–0.5 mg/kg) [124]. Of note, gene therapeutic correction has to be accomplished in the CNS for most LSDs, but not necessarily in FD. Cerebrovascular dysfunction in FD patients resulting in neurological deficits stems largely from stenosis of small vessels and enlargement of large vessels may result in neurological deficits [125].

## 8. A4GALT: Reaction Mechanism and Enzymatic Products

α-1,4-Galactosyltransferase (A4GALT, Gb3 synthase) is the enzyme responsible for the synthesis of Gb3 from LacCer and UDP-galactose catalyzing the formation of an α-glycosidic 1,4 linkage between the anomeric center of the UDP-Gal donor and the LacCer acceptor. This retaining galactosyltransferase is a member of the GT32 family of glycosyltransferases (EC 2.4.1.228, www.cazy.org, accessed on 08-02-2021). In line with many glycosyltransferases, structural and mechanistic information regarding A4GALT is scarce and the exact mechanism is still a matter of debate, making the rational design of inhibitors a very a challenging process. While no crystal structure of A4GALT has been obtained hitherto, the bacterial homologue LgtC (~20% homology) from *Neisseria meningitidis* has shown the presence of a critical carboxylate residue (Asp190) in its active site, potentially situated on the beta face of UDP-Gal and in the vicinity of the lactosylceramide acceptor [126,127]. This carboxylate pointed to the hypothesis of a double displacement mechanism similar to the one employed by retaining glycosidases (Figure 4 and Figure 5A). However, this amino acid is 8.9 Å away from the donor UDP-Gal and a conformational change would be necessary during catalysis to allow an appropriate positioning. The alternative hypothesis invokes an S_N_i-like mechanism (Figure 5B) in which both the incoming nucleophile and leaving UDP group find occupancy in the enzyme active site at the same time [128,129]. Structural studies also showed the presence of a Mn^2+^ cation within the active site of the enzyme. This metal ion interacts with an Asp-X-Asp (DXD) motif and with the diphosphate leaving group of the UDP-Gal donor. Coordination of Mn^2+^ to the diphosphate leaving group assists the departure of the leaving group (UDP) by stabilizing the negative charge [127]. Unveiling the A4GALT mechanism is of great interest for FD, for which clinical targeting is hampered by a complete lack of effective inhibitors.

Current treatment of FD focused on restoring α-GalA activity through ERT or PCT has shown, as previously discussed, limited clinical efficacy. An attractive therapeutic alternative would be the use of SRT which, for instance, has predominantly been successful for the treatment of Gaucher disease [130]. FDA-approved miglustat and eliglustat inhibit GCS, thereby reducing glucosylceramide levels. Of note, the reduction in GlcCer levels would also indirectly reduce the amount of Gb3 formed by A4GALT. However, when compared to GBA activity in Gaucher patients, male patients suffering from FD have extremely low to non-existent activity of α-GalA [131], meaning that full inhibition of GCS would be required. Complete inactivation of GCS could bring serious health risks since glucosylceramide is a key intermediate for the synthesis of other glycosphingolipids (GSLs) essential for various cellular processes such as cell signaling, membrane stability and immunogenicity [132].

Selective inhibition of A4GALT would, in principle, not interfere with the synthesis of other related GSLs. A4GALT is responsible for the synthesis of Gb3, also known as CD77 or the P^k^ antigen, and the P1 antigen [133,134]. Both glycosphingolipids, P^k^ and P1, are blood group antigens belonging to the P1PK system. While P^k^ is a highly frequent antigen on red blood cells (over 99.9% of humans), P1 is present only in a small fraction of the population. The P1 antigen is formed through the coupling between neolactotetraosylceramide (paragloboside) and the UDP-gal donor [135]. P^k^ and P1 are both expressed on the surface of human red blood cells. Recently, it has been shown that the p.Q211E variant of A4GALT is also able to synthesize NOR antigens, which are rare glycosphingolipids with a terminal Gal(α1–4)GalNAc moiety present in erythrocytes of patients with NOR polyagglutination syndrome [135,136] (Figure 6A). Importantly, the existence of some individuals with a genetic deficiency in A4GALT without obvious clinical consequences suggests that selective inhibition of this enzyme could be well tolerated by FD patients [137,138].

Gb3 is the main receptor for Shiga toxins which are released by shigella species and Shiga-like toxins produced by certain strains of *Escherichia coli* (*E. coli*) called Shiga-like toxin-producing *E. coli* (STEC), also referred to as verocytotoxin (VT)-producing *E. coli* (VTEC) [139]. The bacteria usually enter the body via contaminated food or water and can cause serious health problems such as hemorrhagic colitis, which can eventually progress towards hemolytic–uremic syndrome (HUS) [140]. The most common toxins are Shiga toxin 1 (stx1) and Shiga toxin 2 (stx2) and both utilize Gb3 as their cell surface receptor with a similar intracellular mechanism of action [141]. Interestingly, similar to A4GALT knockout mice that are insensitive towards Shiga toxins, increased levels of Gb3 in FD mice also protect the mice against Shiga toxins. However, after administration of recombinant human α-GalA and restoring normal Gb3 levels, FD mice became susceptible to the bacterial toxin [142].

Recently, genome-wide CRISPR-Cas9 knockout screens in Shiga toxins revealed that the lysosomal-associated protein transmembrane 4 alpha (LAPTM4A) is a key player in the biosynthesis of Gb3 [143,144], and LAPTM4A knockout cells showed to be resistant towards Shiga toxins by impairing the binding of the toxins to the cell surface due to the lack of Gb3 [143]. However, the absence of LAPTM4A did not affect A4GALT levels or its proper localization with A4GALT in the Golgi complex. Further analysis showed that the second luminal domain of LAPTM4A plays a key role in the interaction with A4GALT and the eventual synthesis of Gb3 [144]. Moreover, it was shown that replacing only the second luminal domain of LAPTM4A in the homologous LAPTM4B also restores Gb3 synthesis [144]. The fact that A4GALT activity in vitro, with an artificial lipid substrate (NBD-LacCer), is not dependent on the presence of LAPTM4A suggests that this protein could be involved in the presentation of the lipid substrate (LacCer) from membranes to the enzyme [144]. However, additional studies towards the structure and function of LAPTM4A are necessary to fully understand its exact role in Gb3 metabolism. Importantly, due to the relation between LAPTM4A and A4GALT and the resulting influence on cellular Gb3 levels, LAPTM4A or their protein–protein interaction could also be a potential therapeutic target for the treatment of FD.

## 9. A4GALT Inhibitors and Future Directions

Lowering Gb3 levels remains an important, though challenging, therapeutic strategy for FD. Despite the important role of GTs in various biochemical processes, these enzymes have been relatively unexplored compared to GHs and their respective inhibitors. In general, GT inhibitors are predominantly developed via a rational design approach based on donor or acceptor analogues or by high-throughput screening (HTS) [145]. The first and most logical therapeutic target for SRT in FD is A4GALT. However, no inhibitors have seen the light and only adamantyl-functionalized galactosylceramide (adaGalCer) has shown competition with the LacCer substrate and inhibits Gb3 synthesis in cells [146]. One main reason for the slower development of A4GALT inhibitors may be the lack of structural and mechanistic information concerning the enzyme.

Adamantyl galactosylceramide (adaGalCer) and glucosylceramide (adaGlcCer) are A4GALT acceptor analogues with a modified ceramide fatty acid tail functionalized with an adamantane (Figure 6B). These compounds alter GSL metabolism. In particular, adaGalCer acts as a substrate for A4GALT and is able to lower Gb3 levels at an IC_50_ concentration of 40 μM in FD cells [146]. Of note, enzymatic galactosylation of the inhibitor results in the formation of the adaGb2 product with unknown physiological consequences. However, this artificial more apolar metabolite was 10-fold more effectively excreted to the medium than Gb3 in cells, suggesting a better elimination and a potential solution to Gb3 accumulation. On the other hand, adaGlcCer is converted to adaLacCer and inhibits LacCer synthesis.

A second rational strategy for development of A4GALT inhibitors could be the synthesis of UDP-Gal mimics. Following the success of GH inhibitors, fluorinated donors with a fluorine at their C2 or C5 position functionalized with a UDP group at the anomeric position have been developed as slow inhibitors of retaining glycosyltransferases [147,148,149]. UDP-carba-Gal analogues, in which the pyranose oxygen atom is replaced by a carbon atom, have also been developed as GT inhibitors and, in general, are very stable competitive inhibitors [150]. In addition, iminosugar donors have been described as GT inhibitors and show electronic and structural similarity by mimicking the positive charge in the oxocarbenium ion transition state [151]. C-glycosides, in which the exocyclic oxygen is replaced by a carbon, also function as donor mimics without being prone to hydrolysis by GHs [152], and a C1-C2 alkene-based analogue conformationally mimicking the oxocarbenium ion transition state resulted in a low-affinity β-galactosyltransferase inhibitor [153]. Different modifications at the nucleotide base have been exploited as well [154]. For instance, attachment of a 5-formylthien-2-yl group to the 5′ position of the base resulted in a nanomolar inhibitor of several different GalTs by blocking the movement of a key mobile loop in the enzyme structure [155]. Of note, the selectivity of UDP-based inhibitors is questionable since a particular UDP-sugar donor functions as a substrate for multiple GTs.

Importantly, the identification of new A4GALT inhibitors could provide important structural and mechanistic insights. Proteomics combined with crystallographic studies using mechanism-based inhibitors could shed some light on the presence or absence of a covalent enzyme inhibitor intermediate and determine if the enzyme actually catalyzes the glycosylation via a double displacement mechanism or a S_N_i-like concerted front-face mechanism. For these inhibitors to become a reality, future research towards effective A4GALT biochemical assays and new HTS methodologies to potentiate the discovery of new binders, together with A4GALT crystallographic studies, appears essential.

## 10. Concluding Remarks

The detailed knowledge on the molecular basis of FD has not yet resulted in a very effective treatment. Ongoing research on modified enzymes without immunological responses and the design of new treatment modalities, such as gene therapy, enzyme stabilizers or SRT targeting A4GALT in a selective and controlled manner, hold promise to reach major improvements in this direction.

## Figures and Tables

**Figure 1 biomolecules-11-00271-f001:**
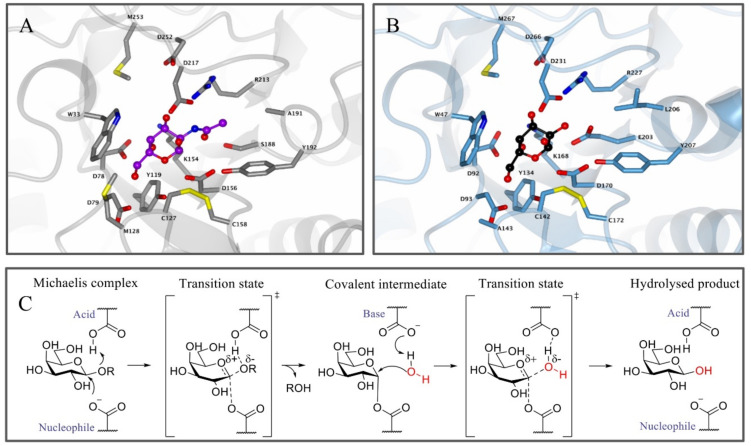
Enzyme structure and reaction mechanism of α-N-acetylgalactosaminidase (α-GalB) and α-galactosidase A (α-GalA). (**A**) Active site of α-GalB (gray) with GalNAc (blue) bound in the pocket. (**B**) Active site of α-GalA (blue) with galactose (black). Larger C-2 substituents cannot be accommodated in α-GalA due to the presence of residues L206 and E203. Structures were obtained from the Protein Data Bank (PDB) IDs 3H55, 3H54 or 3GXP and visualized using CCP4MG. (**C**) Koshland double displacement mechanism of retaining α-GalA.

**Figure 2 biomolecules-11-00271-f002:**
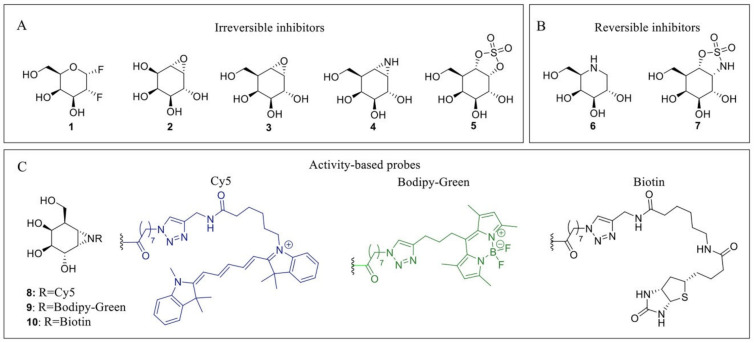
α-GalA inhibitors and activity-based probes (ABPs). (**A**) Irreversible inhibitors: 2-deoxy-2-fluoro-D-galactosyl fluoride **1**, conduritol C **2**, cyclophellitol epoxide **3**, cyclophellitol aziridine **4** and cyclosulfate **5**. (**B**) Reversible inhibitors: Gal-DNJ **6**; cyclosulfamidate **7**. (**C**) ABPs: Cy5 probe **8** (blue), Bodipy-green probe **9** (green) and biotinylated probe **10** (black).

**Figure 3 biomolecules-11-00271-f003:**
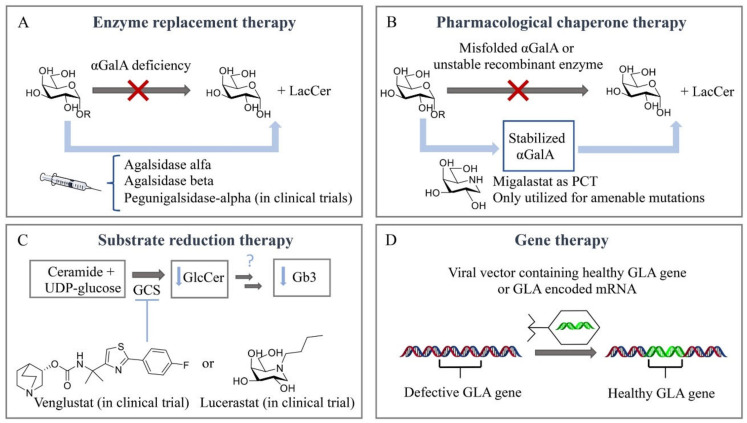
Therapeutic strategies for treatment of Fabry disease. (**A**) Enzyme replacement therapy (ERT). (**B**) Pharmacological chaperone therapy (PCT). (**C**) Substrate reduction therapy (SRT). (**D**) Gene therapy.

**Figure 4 biomolecules-11-00271-f004:**
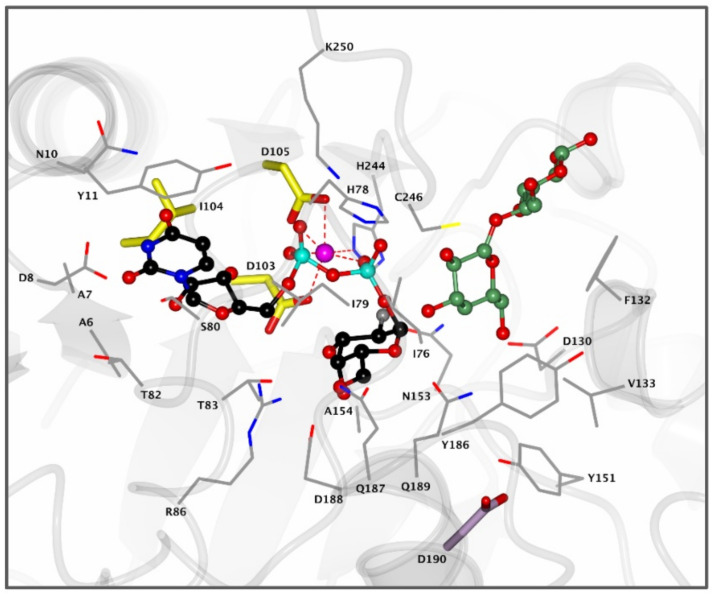
Active site of bacterial A4GALT homologue LgtC. Enzyme active site containing donor analogue UDP-2FGal (black) and acceptor analogue 4′-deoxylactose (green) bound in the pocket. The Mn^2+^ cation (pink) coordinates with the diphosphate group of UDP-Gal and the DXD motif (yellow) to assist in catalysis. Residue D190 (purple) is positioned 8.9 Å away from the UDP-Gal donor and is visualized for clarity. Structure was obtained from the Protein Data Bank (PDB) ID 1GA8 and visualized using CCP4MG.

**Figure 5 biomolecules-11-00271-f005:**
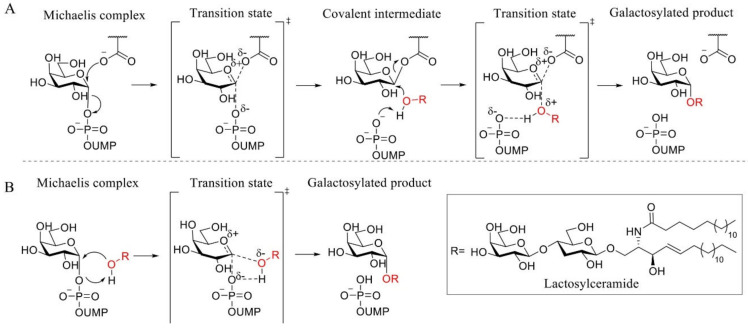
Proposed A4GALT mechanisms. (**A**) Koshland double displacement mechanism of retaining glycosyltransferases (GTs). (**B**) Front-face (S_N_i-like) mechanism of retaining GTs.

**Figure 6 biomolecules-11-00271-f006:**
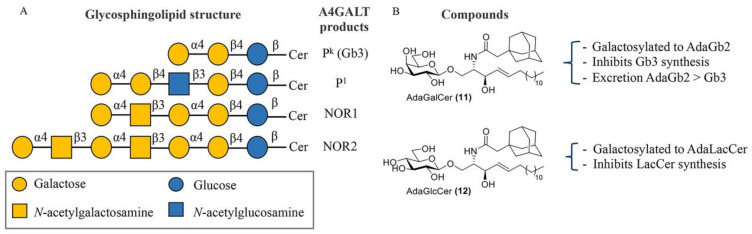
A4GALT glycosphingolipid (GSL) products and Gb3 modulators. (**A**) Structures of glycosphingolipids produced by A4GALT. (**B**) AdaGalCer **11** and AdaGlcCer **12** and their effect on GSL production.

## Data Availability

Not applicable.

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
