# Peer review of "Fabry Disease: Molecular Basis, Pathophysiology, Diagnostics and Potential Therapeutic Directions"

_biomolecules, 2021, doi:10.3390/biom11020271_

Round 1

Reviewer 1 Report

This will be a useful review of Fabry therapies and treatments both to those familiar with the field and those new to the field I have a few suggestions for the authors. 

  1. The review assumes that the reader is familiar with treatment rationale in the LSD field in general. It might be useful if a short section was inserted explaining the rationale of ERT, PCT, SRT and gene/RNA therapies. This in particular would br invaluable help those new to the field.
  2. 2. Line 53 Clinical manifestation of FD. Again a short sentence or two on the genetic inheritance of FD may be helpful to some. X-linkage? Lionisation?
  3. Line 318 Again misfolding and mechanism of chaperone activity might be a little more fully explained.
  4. Line 333 "joint" rather than "jointly" 
  5. Figure 3 Needs linking to suggestion 1.
  6. Page 10 line 391. It might be worth mentioning here that gene therapy has to reach the CNS in sufficient quantity for most LSDs, not Fabry.
  7. Line 505 there is a mis-wording here should it be " mechanistic information concerning the enzyme'?

Reviewer 2 Report

The review article "Fabry disease: molecular basis, pathophysiology, diagnostics, and potential therapeutic directions" described the pathology of Fabry disease and therapeutic approaches. Further addresses the biochemical basis of Fabry pathology.

The text is well written, diagnostic and therapeutic directions are details and accurate.  In regard to the molecular pathology, the manuscript falls a bit not updated and could use some revamping.

 I would like to mention some further points of criticism, which should be worked on before the article is suitable for publication.

Main points:

The text contains old literature references for numerous statements. For example:

  • P106-P112: Several recent studies (2017-2020) and reviews addressed the mechanism of cellular dysfunction, the effect of autophagy-lysosomal malfunction, and mitochondrial function in Fabry pathology, including the patient's primary cells or tissues.  
  • Studies uptake of the recombinant enzyme, will be good to add more recent publications.
  • P322-323: Migalastat (FDA approval in 2018 mean in USA?)
  • Figure 3C: GlcCer inhibition directly linked to Gb1 inhibition and did not directly relate to Gb3 inhibition.
